



# Grand Challenges in the Digitalisation of Wind Energy

Andrew Clifton[1,*], Sarah Barber[2,*], Andrew Bray[3,*], Peter Enevoldsen[4,*], Jason Fields[5,*], Anna Maria Sempreviva[6,*], Lindy Williams[7,*], Julian Quick[8], Mike Purdue[9], Philip Totaro[10], and Yu Ding[11]

[1]Stuttgart Wind Energy at the Institute of Aircraft Design, University of Stuttgart, Germany
[2]Eastern Switzerland University of Applied Sciences, Oberseestrasse 10, 8640 Rapperswil, Switzerland
[3]MXV Ventures, Oakland, California, USA
[4]Centre for Energy Technologies, Aarhus University, Denmark
[5]National Renewable Energy Laboratory, Golden, CO, USA
[6]Technical University of Denmark, DTU. Department of Wind Energy Risø Campus Frederiksborgvej 399, 4000 Roskilde, Denmark
[7]National Renewable Energy Laboratory, Golden, CO, USA
[8]University of Colorado, Boulder, CO, USA
[9]NRG Sytems, Hinesburg, VT, USA
[10]IntelStor LLC, Houston, TX, USA
[11]Department of Industrial and Systems Engineering, Texas A&M University, College Station, TX, USA
[*]These authors contributed equally to this work.

**Correspondence:** Andrew Clifton (andy.clifton@enviconnect.de)

**Abstract.** The availability of large amounts of data is starting to impact how the wind energy community works. From turbine design to plant layout, construction, commissioning, and maintenance and operations, new processes and business models are springing up. This is the process of digitalisation, and it promises improved efficiency and greater insight, ultimately leading to increased energy capture and significant savings for wind plant operators, thus reducing the levelized cost of energy.

Digitalisation is also impacting research, where it is both easing and speeding up collaboration, as well as making research results more accessible. This is the basis for innovations that can be taken up by end users. But digitalisation faces barriers. This paper uses a literature survey and the results from an expert elicitation to identify three common industry-wide barriers to the digitalisation of wind energy. Comparison with other networked industries and past and ongoing initiatives to foster digitalisation show that these barriers can only be overcome by wide-reaching strategic efforts, and so we see these as "Grand

Challenges" in the digitalisation of wind energy. They are, first, the need to create reusable data frameworks; secondly, the need to connect people to data to foster innovation; and finally, the need to enable collaboration and competition between organisations. The Grand Challenges thus include a mix of technical and cultural aspects that will need collaboration between businesses, academia, and government to solve. Working to mitigate them is the beginning of a dynamic process that will position wind energy as an essential part of a global clean energy future.



## 1 Introduction

The global green transition is the process of changing our technological base and consumption patterns to reduce and ultimately reverse anthropogenic climate change. An integral part of the green transition is the adoption of very high levels of renewable energy sources, such as wind energy (Jacobson, 2017).

In an article in *Science*, Veers et al. (2019) discussed how to secure a competitive wind energy price in the energy market, and
positioning wind as one of the primary sources of the world's electricity generation by 2050. Their paper, "Grand challenges in the science of wind energy," identified three high-level Grand Challenges:

1. Improved understanding of atmospheric and wind power plant flow physics

2. Aerodynamics, structural dynamics, and offshore wind hydrodynamics of enlarged wind turbines, and

3. Systems science for the integration of wind power plants into the future electricity grid.

These Grand Challenges are similar to the long-term research challenges in wind energy identified by van Kuik et al. (2016). However, a response to the Veers article pointed out that there were other important Grand Challenges related to environmental and ecological impacts, as well as social, "territorial and institutional dimensions",[1] and there have been other discussions since about what the Grand Challenges really are and how the international wind energy community is responding to them (e.g., Wood, 2020).

### 1.1 The role of digital technologies in addressing the Grand Challenges

Several things are clear from the "Grand Challenges" paper and other discussions of the future direction for wind energy. Firstly, the challenges cover multiple scales in space and time. Secondly, they are interrelated and cannot easily be separated. And, thirdly, addressing them will need an interdisciplinary research and development approach.

These three points suggest the need to combine and use data from many different sources and disciplines. Accordingly,
multiscale and interdisciplinary data will play a fundamental role in addressing each of the wind energy Grand Challenges, fuelling the transition to a renewable energy system. The opportunities are potentially considerable; as a U.S. Chamber of Commerce Foundation research report noted in 2014,

> "Regardless of what form it takes, data tells a story. It can identify cost savings and efficiencies, new connections
> and opportunities, and an improved understanding of the past to shape a better future. It also provides the details
> necessary to allow us to make more informed decisions about the next step we want to take." (Heinz et al., 2014)

In this paper, we explore the possibilities offered to the wind energy sector by the availability of continuously developing digital technologies—such as storage, connectivity, computational power, data management, data science tools, digital twins, and many others—to exploit the ever-increasing amount of data. These lead to opportunities for efficiency, innovation, and

---

[1]See https://science.sciencemag.org/content/366/6464/eaau2027/tab-e-letters.





entrepreneurialism. Together, these are many of the essential aspects of digitalisation, which we define as the organisational
and industry-wide use of data and digital technologies to improve efficiency, create insights, and develop products and services.
In short, digitalisation is doing new, value-driven things with data and tools generated in digital form by the digitisation process.

Digitalisation is foreseen as a lever for the global green transition through empowering information and communication
technologies to increase resource efficiency and to accelerate research and development (Fernández-Portillo et al., 2019; Cas-
tro et al., 2021). This process is already happening in many sectors; the manufacturing sector has seen that significant business
advantages can be obtained by combining automation and data exchange with manufacturing technologies, which is itself a
form of digitalisation. This trend is known as "Industry 4.0" and follows previous sector-wide manufacturing industry transi-
tions including the use of steam power, electrification, and the introduction of computing. There are reasons to expect similar
trends in energy; in 2017, a report by the United Nations Industrial Development Organization (UNIDO) noted

> "The sustainable energy transition and Industry 4.0 share important characteristics: both are highly influenced by
> technological innovations, dependent on the development of new suitable infrastructures and regulations as well
> as are potential enablers for new business models." (Nagasawa et al., 2017).

Historically, such technological transitions are driven by a combination of curiosity, need, and cultural pressures, and occur
when a disruptive technological innovation emerges from a niche (Geels, 2002). Strong connections between the niches and
the broader landscape help to create a pull for new technologies that are relevant to users, helping increase adoption. Often
the process of adoption is transformational for both the new technology and the market, and while some new innovations fail,
others go on to become part of the technological landscape. The process of innovation therefore does not happen at the same
time across a sector, but is instead uneven and happens opportunistically, and probably where the barriers are lowest.

## 1.2 Opportunities for the use of data

Data has many applications in the wind energy industry and is already helping to improve efficiency and create new business
models. For example:

– Data about a wind energy development site's characteristics are fundamental requirements for many different aspects of
site design and optimisation (see e.g., Brower, 2011; Clifton et al., 2016; Floors et al., 2018). Experience from operating
sites is used to calibrate and refine the design process and improve new sites.

– Large amounts of data are collected on wind turbines by sensors installed by the original equipment manufacturer
(OEM). Coupling sensor data with models of the turbine allows the creation of "digital twins" of wind turbines (Rinker
et al., 2018). These software representations of hardware are now available as commercial services and are frequently
used for tracking turbine fatigue loading and remaining lifetimes (Branlard et al., 2020). Research is ongoing on ways
to use this data to make intelligent control decisions (Pettas et al., 2018). Similarly, data-based condition monitoring of
wind turbines is common, with many companies using statistical analysis or machine learning to provide early warning
of component failure (Colone et al., 2019). Third-party sensors are also gradually being used on wind turbines; for





example, data from forward-looking wind lidar are used to inform control decisions such as blade angles or generator torque, potentially reducing loads (Schlipf et al., 2018).

- Wind energy needs to be integrated into our electrical supply. This means that wind turbines must respond to the grid at timescales on the order of seconds or less. Traditional thermally-powered electricity generators automatically change their power output in response to changes in grid frequency. Wind turbines lack this response. Instead, their output can be regulated by an external control signal or by monitoring the grid locally (Denholm et al., 2020). Power forecasts driven by real-time observations also help enable the integration of wind turbines (Würth et al., 2019).

- Available data is being used to understand the dynamics of wind fields as well as turbine reliability, evaluation, assessment, and performance (Ding, 2019).

- Existing processes such as wind resource assessment are gradually being encoded so they rely less on human inputs and thus run faster or more efficiently (Holleran et al., 2022). This should reduce development times or allow optimisation, reducing the cost of energy.

- Complex issues such as social acceptance and environmental impact are starting to be targeted (Permien and Enevoldsen, 2019). This has resulted in such ideas as using virtual or augmented reality to preview a wind farm before it is built or collecting real-time feedback from local residents about noise or flicker (Cranmer et al., 2020). Also, wind turbines can be linked with bird- or bat detectors to trigger avian-friendly operating modes (Salkanović et al., 2020); these may become even more important if such externalities are assigned a price in future.

Digitalisation in the wind energy industry is progressing. But so far it tends to be vertical or siloed within an organisation or takes place directly between one organisation and another. This internal focus is not unique to the wind energy industry; Branca (2019) found that initial digitalisation efforts in the metals processing industry were often internal to an organisation and used to optimise internal processes, manufacturing, or products.

However, the wind energy industry is highly interconnected both physically and in terms of the interdisciplinary science and engineering that underpins it. This paper directly addresses the challenges in wind energy digitalisation, but additional technology, such as cybersecurity, are vital to enabling the future described. This, in combination with the innovation style of the leading industry actors (Sovacool and Enevoldsen, 2015), suggests that collaboration may be an important aspect of the future of digitalisation of wind. Steps have been taken to initiate collaboration, for example by sharing (some) research data through online platforms. By comparison, very little operational data is freely available, and the wind energy industry acknowledges its own reticence to share data (e.g., Lee et al., 2020).

### 1.3 The potential for digitalisation to reduce the cost of energy

Access to wind turbine data can be leveraged to improve the wind turbine and wind plant design process, resulting in smaller uncertainties in loads, lifetimes, and power production. In turn, this saves on raw materials, saves on project financing costs, and increases stakeholder confidence. Studies estimate that this might allow a 1% capital expenditure (CAPEX) saving. As a




large floating wind farm may be a multibillion-dollar investment (Ghigo et al., 2020), this could reduce CAPEX by tens of millions of dollars.[2] Without cultural changes that allow access to past data and technical changes that enable new approaches

to designing and operating wind turbines, this saving would simply not be possible.

Operational expenses (OPEX) can also be reduced by coupling condition monitoring with intelligent maintenance scheduling, rather than simply responding to failures as they happen, potentially saving several percent of OPEX. One vendor of improved predictive maintenance using data and machine learning estimates that

> "... up to 30% of the levelised cost per kWh produced over the lifetime of a turbine can be attributed to Operation
> and Maintenance..."

They then claim that

> "Using [brand name of a digital service], one can expect a reduction in replacement and labor costs. Assuming a
> 20% reduction in the repair and maintenance portion of O&M costs, this would translate into annual cost savings
> of $11,383 for a 2.5-MW turbine and $34,148 for a 7.5-MW turbine." [3]

Scaled to a 100-turbine facility, this offers over $3 million in savings. But, the assumption of 20% savings may be optimistic; a wind energy industry market research company interviewed for this paper, noted

> "... an average reduction of 11% of the operations and maintenance cost for operational assets when utilising
> predictive maintenance facilitated by data analytics versus conventional reactive maintenance practices."

In both cases, it is not clear how much the digital services cost, so the net savings are unknown. There are also some
challenges to adopting predictive maintenance systems. One is the total cost. Reducing the life cycle costs of after-market sensors will likely make it easier to justify adding sensors to already-operating turbines, while the implementation of "plug-and-play" standardised virtual environments and the development of innovation marketplaces for data science applications will reduce the cost of providing services.

Digitalisation can also help increase the amount of energy delivered by a wind energy facility, and thus increase income.
It can increase the efficiency of a turbine by detecting blade soiling, damage, or erosion; can enable wind-plant level control strategies that reduce wake interaction and reduce energy losses; and access to data can help when planning regional and national transmission and distribution to help avoid curtailment. A wind energy industry market research company interviewed for this paper, noted

> "The additional annual energy production (AEP) from selective power uprating has ... resulted in approximately
> 2.4% additional power compared to the same average without the digital service being provided."

---

[2]Estimated CAPEX for offshore wind energy systems in 2014 were around €3.5 - €4.5 Million per MW (Myhr et al., 2014). A 1% CAPEX saving for a gigawatt-scale plant is therefore equivalent to €35 - €45 Million. Even with up to 50% reductions in future offshore wind CAPEX due to industry learning, this would be a valuable saving.

[3]From "Maintenance 4.0 in wind farms: Bringing smart analytics to clean energy". https://industrial-ai.skf.com/maintenance-4-0-in-wind-farms. Accessed May 2021.



**Table 1.** Definitions for digitisation, digitalisation, digital transformation, and digital businesses

| | |
|---|---|
| Digitisation | the process of converting information into digital signals, enabling digitalisation (Rachinger et al., 2019) |
| Digitalisation | the organisational and industry-wide use of data and digital technologies to improve efficiency, create insights, and develop products and services |
| Digital transformation | the holistic process which integrates and adopts digitalisation on societal levels (Mergel et al., 2019) |
| Digital businesses | businesses that incorporate digitalisation into their activities and derive a significant part of their turnover from digital products and services |

While significant, these values are below the double-digit gains promised by some digital service companies, and it is possible that over-selling has slowed the pace of industry adoption of digital solutions.

Additionally, digitalisation of wind energy potentially increases the value of the energy. This often depends on market structure. One way to do this in open electricity markets is to store energy, and then sell it when it is more valuable. Automated energy trading or human-controlled energy trading with decision support tools is common in some markets but requires accessible data and an open exchange. Improved weather and demand forecasts are already being used to reduce imbalance charges and penalties, but this also requires specific market structures. Weather forecasts can be improved by data sharing by multiple organisations in a region, but this approach will need secure data sharing to be effective.

These examples are technology-focused cases of digitalisation, yet one must differentiate between digitalisation and a) digitisation, which is the conversion of analogue data and thereby the enabler of digitalisation (Rachinger et al., 2019), and b) digital transformation being the definition of the holistic process which integrates and adopts digitalisation on societal levels (Mergel et al., 2019). The formal definitions used in this paper are given in Table 1.

These examples show that the barriers to new business models are often not scientific or technical—because the underlying technologies generally already exist—but might be related to concerns around competition, are driven by commercial interest, or related to the market they operate in. The examples also introduce some of the important stakeholders in the digitalisation of a commercial wind energy facility. They include investors, project managers, technologists, site workers, and many others. Research projects may have different stakeholders, for example, researchers and funding agencies. Each stakeholder group has different perspectives, which further demonstrates how digitalisation is potentially as much a cultural challenge as a technical challenge.

## 1.4 How digitalisation might affect how we work

To help the reader understand how digitalisation could impact the daily life of someone in the wind energy industry, we have combined some of the current technology and business trends - digital twins, predictive maintenance, drones, expert systems, and many others - into a snapshot of a few hours in the life of a colleague in the year 2030. While this is speculative, it helps see some of the impacts, opportunities, and challenges that digitalisation could bring.



Alice was involved in the initial engineering design of an 800-MW floating offshore wind plant that was built in the mid-2020s. She led the commissioning and has been managing it ever since. She has a simple remit from the project owners: to keep everyone safe and make sure that the plant meets its financial targets. Her story follows.

*Alice's phone pinged, and her screen flashed with a warning: one of the new 20-MW floaters wasn't happy. They'd had a few teething problems with the power-to-fuel plant in one of the spars since they'd installed them off the coast a few years ago.*

*The plant was run by an intelligent supervisory system trained on data from an onshore pilot plant. It ran just fine most of the time, and maintenance requests were added to the site engineer's dockets automatically, but sometimes a human touch was still needed. Alice's screen greeted her with a few pop-ups each morning. Receiving those simple summaries of what had happened and why, and what could be done, really helped. All she had to do was tap a few buttons and move on to the next thing on her agenda.*

*Now that they trusted the automated monitoring, the maintenance folks had added automatic inspection and repair drone deployment to take advantage of good weather. Digital twins kept track of what was going on, and they were so accurate in predicting maintenance needs that some of the crew jokingly called them Digital Clairvoyants.*

    *A few old-timers had grumbled when the system came online. They'd worried about being put out of a job by machines. Instead, they were now able to work on complex jobs where a drone or robot crawler couldn't go or couldn't figure out what*

*to do. Now they could focus on the challenging jobs instead of taking on every job. This change also gave them the breathing room to focus even more on safety. And, as more plants came online up and down the coast and started sharing their weather data, the forecasts were improving too—no more getting caught out in high seas or freezing rain anymore.*

    *The design team had seen it coming. Back in 2022, when the first prototype floating systems were commissioned, they'd invested in lots of sensors and engineering models for everything from the sea state to loads and fatigue. They let the world*

*know that data was there and asked for help analysing it. The data and services marketplace that had grown was now benefiting everyone. Better-designed turbines and more reliable energy were reducing costs across the board. In contrast, some of their competitors had foregone installing sensors to save a few dollars. But now they had no idea what had caused some of the problems they had seen, or why their turbines weren't performing as expected.*

    *Alice and her group, on the other hand, could leverage their treasure trove of data to understand what their turbines were*

*experiencing and what the turbines needed. That data let them collaborate and innovate. And now, unlike their competition, they were seeing big savings - tens of millions of dollars at their site alone. They had larger turbines than anyone else, but cheaper and lighter. And the turbines were smart too, providing grid support services and not just pushing electricity to the grid.*

    *The turbines could also power synthetic fuel production when the grid had more electricity than it needed. Now, the model*

*estimates were telling them that they had sold more synthetic fuel futures to the local shipping lines than they were going to have available to supply in the next month. The updated model was only adjusted 5% from the original model a month ago, but the adjustment from the updated sensor information would earn them millions of dollars more over the years. With a wave of a hand, she sold back a future with a secure transaction on the open exchange to an unknown counterparty to banish the warning.*



Although Alice's experience the operations phase of the plant, digitalisation will benefit all stages of the wind plant life cycle, from the design phase through construction and commissioning, to operations and maintenance (Figure 1). And these are not fiction or wishful thinking; many of the opportunities seen in the story have already been demonstrated and are approaching commercialisation or are already in daily use.

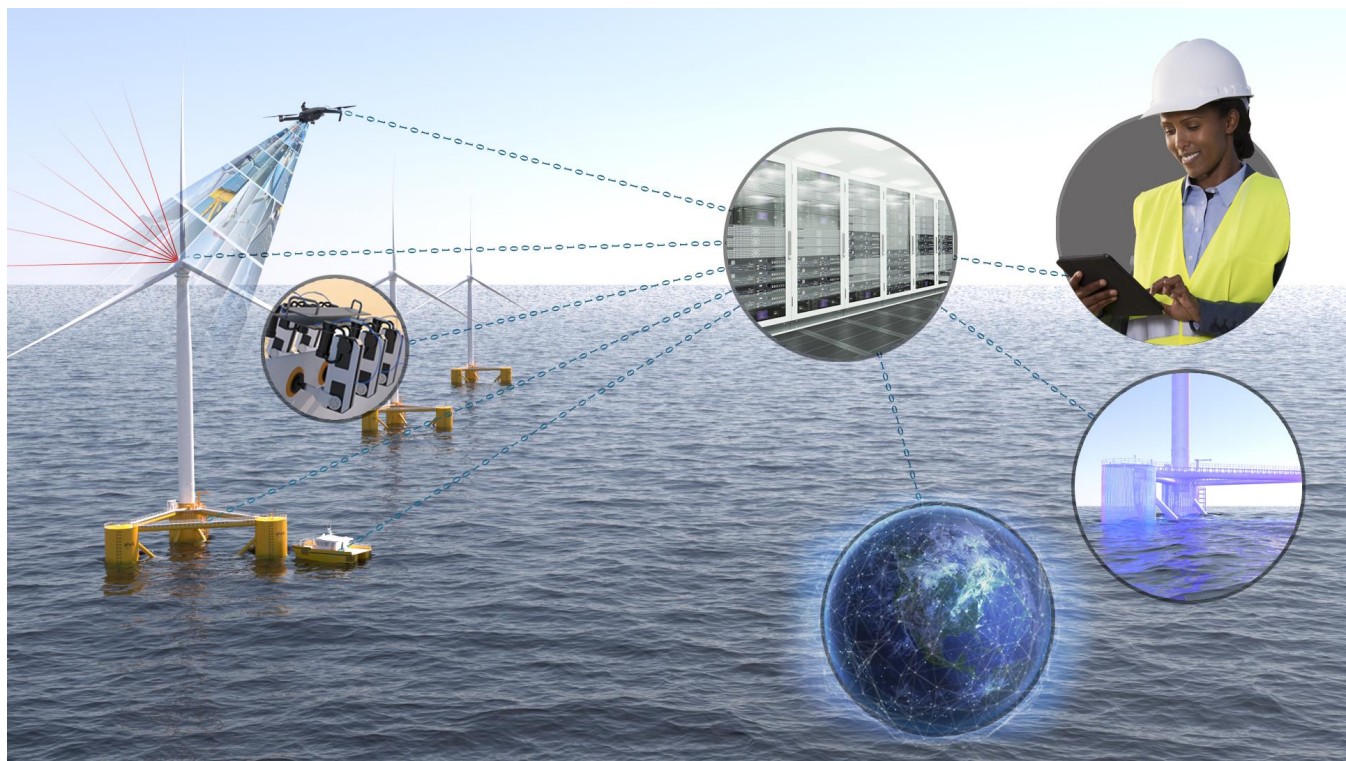

**Figure 1.** Digitalisation in action. In this future floating wind energy plant, digitalisation enables a plant manager to take data-based decisions in real-time, increasing safety and reducing the cost of energy. Image credit: NREL graphics team

## 1.5   A research-driven assessment of the Grand Challenges for digitalisation

An initial literature review revealed that few studies have examined the consequences, level, and trends of digitalisation in the wind energy industry. In response to this apparent lack of research, we report here the results of a literature survey and data collection activities to understand how digitalisation is progressing, how it might lead to fundamental changes in the wind energy industry, and the issues that would be faced *en route*. Our goal is to identify sector-level Grand Challenges for the digitalisation of wind energy. We consider these to be things that must be done for the wind energy sector to digitalise; that impact

many different stakeholders; and require coordinated effort to solve. They might also be issues that are seen again and again in different markets, such that they are the same challenge but with different solutions. Identifying the Grand Challenges therefore requires combining an understanding of the process of digital transformation with many different stakeholder perspectives.





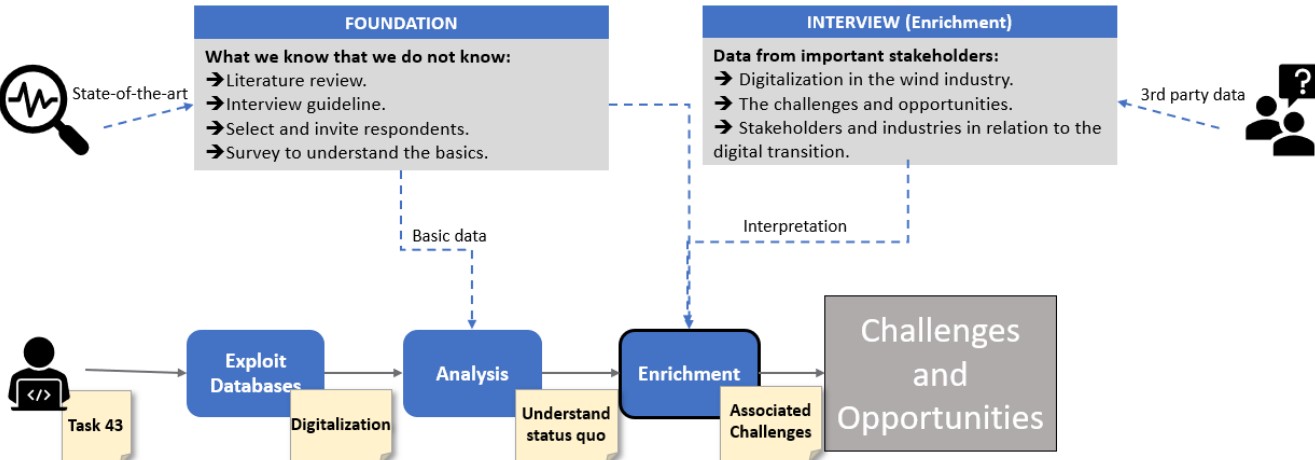

**Figure 2.** Data collection, processing, and analysis were used to help identify the Grand Challenges for digitalisation of wind energy

This paper is targeted at policy advisers, funding agencies, research managers, and others involved with technology transfer at a strategic level. We expect that it will also be of interest to technologists and researchers as it may provide insight into future
research directions.

Section 2 presents the results of an initial expert elicitation that informed this study and the identification of the Grand Challenges. Section 3 describes the process of digital transformation in the context of wind energy. Section 4 provides lessons learned from other high technology networked industries. Based on this, we carry out a more specific expert interview series and use the results to identify the Grand Challenges for the digitalisation of wind energy in Section 5. Section 6 presents our
conclusions.

## 2 Research Methods and Materials

The Grand Challenges in the science of wind energy were identified through a process of expert elicitations and synthesis (Veers et al., 2019; Dykes et al., 2019). In this paper we follow a similar process (Figure 2). First, a literature survey coupled with expert elicitations were used to understand the current state of digitalisation in the wind energy sector and actors' per-
ceived challenges. We then refined some of this information through further interviews. The results of the literature survey are presented in this section, while the results of the literature survey are presented in context in Section 3 and Section 4. This information is combined in Section 5 to identify the Grand Challenges in the digitalisation of wind energy.

### 2.1 Data collection

The data presented and applied in this study relies on an empirical collection via a survey and semi-structured interviews with
digitalisation experts, innovation frontrunners, and energy industry market leaders. Other community surveys have also been



carried out recently, and their published results have been considered when preparing this document. For example, in 2019, a survey was carried out to explore the expectations and priorities of wind energy operations and maintenance experts (Berkhout et al., 2020). The data collection and subsequent processing and analysis followed the process exemplified in Figure 2.

## 2.2 Survey

Members of International Energy Agency (IEA) Wind Task 43 and the wind energy industry were surveyed to determine trends of digitalisation in wind energy. The 102 respondents provided 68 different open answers to four questions, which were analysed and used as background knowledge in this study. A specific emphasis was targeted towards critical trends where data management and data sharing were found to be the most critical trends in digitalisation today, while smart energy and robotics were listed as the topics of the future. The output is interesting, as respondents decided to focus on the prerequisite of digital

activities as the critical trend of today, before diving into actual digitalisation activities.

## 2.3 Expert interviews

In parallel to the survey, semi-structured interviews were conducted with 44 digitalisation experts, mainly from the wind energy sector. The respondents were not involved in the initial survey. The interviews followed a guideline based on the initial literature review and IEA Wind Task 43's activities. They were organised to gather insights from a broad range of wind energy

life cycle area experts and digitalisation technology area experts. The Task 43 team interviewed experts in many wind energy life cycle areas, including turbine design and manufacturing, development, construction, operations and maintenance, as well as life assessment and decommissioning. Several technology area experts in data collection, data management, data analytics, and automation were also interviewed (Figure 3).

**Figure 3.** The interviewees' job titles. IEA Wind Task 43 interviewed 44 experts about wind energy digitalisation to help identify the grand challenges. The experts come from across the wind energy industry and have a wide range of roles, as evidenced by the many different job titles that they use. The size of the blocks shows the relative frequency of each title. Colors are arbitrary.





Beyond providing a general understanding of digitalisation efforts in the wind energy sector, we categorised the output of
the interviews into separate challenges (Figure 4) and opportunities (Figure 5).

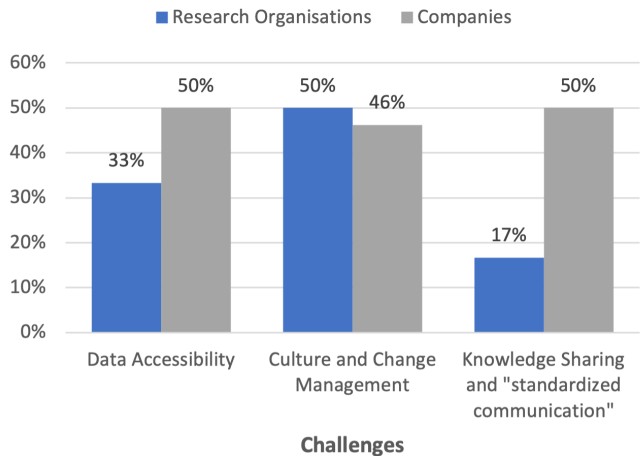

**Figure 4.** The top three digitalisation challenges in the wind energy sector. The result is presented and analysed by stakeholders in academia, including research organisations, and in the private sector, respectively.

Figure 4 indicates that companies perceive data accessibility and knowledge sharing as greater challenges than culture and change management, when compared to research organisations. Nevertheless, consensus was established among the two stakeholder groups on the three dominant challenges.

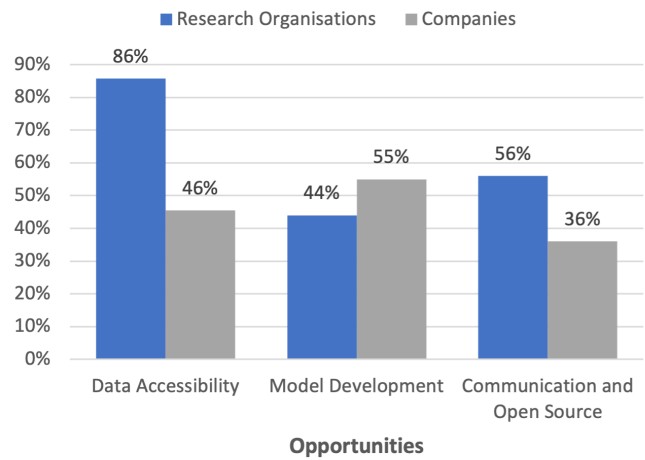

**Figure 5.** The top three digitalisation opportunities in the wind energy sector for different types of organisations





Figure 5 highlights that the two stakeholder groups also consider data accessibility an opportunity. Model development and
"communication and open source" were also considered opportunities; research organisations see the greatest potential in the
latter.

The interview outputs were analysed and divided into industry and research-focused stakeholder groups, which again were
represented by eight companies and five research institutions. Table 2 explains the top challenges and opportunities.

**Table 2.** The top challenges and opportunities identified through the first interview series

| Challenge or opportunity | Explanation |
| --- | --- |
| Culture and Change Management | The challenges associated with the digital transformation requires company alignment, personal motivation, coordination of the organisation with its people, and tangible use cases and outcomes. |
| Data Accessibility | The challenge addresses both the lack of data accessibility and sharing in the vertical and horizontal value chain, as well as the opportunity of obtaining more, and better quality, data than in the past. |
| Knowledge Sharing | The challenge of standardised structures for interorganisational sharing of models, best practices, collaboration beyond data, etc. |
| Model and Algorithm Development | The opportunity to describe phenomena and replicate such methods applied. |
| Communication and Open Source | The opportunity of open-source data, codes, and models, which simultaneously enables common communication and resource sharing among organisations. |

The identified challenges will be analysed following a retroductive approach in the following sections where the require-
ments, process, and opportunities related to the digital transformation of the wind energy industry will be discussed based
on the output from the survey and interviews. Then, the outlook on digital opportunities will be analysed in relation to the
digitalisation experienced in other industries. The empirical data collection, literature review, and analyses then identify the
Grand Challenges in the digitalisation of the wind industry.

## 3   Understanding the Digital Transformation

The survey data and interview findings described in the previous sections led us to further investigate the meaning of the digital
transformation. This work provided us with a framework for how the digitalisation process generates the innovation that drives
the digital technological transition. This transition occurs by taking advantage of the availability of digitised data and novel
data science tools, thus leading to new products, processes, and business models (Figure 6), and transforming the technological
landscape.

In this section, we consider each of the components of the digital transformation. We describe some of the benefits and
challenges associated with each of them and where we see a bottleneck in the process. Finally, we summarise the key aspects
of the digital transformation most relevant for the wind energy industry, which are used towards defining the Grand Challenges
in Section 5.





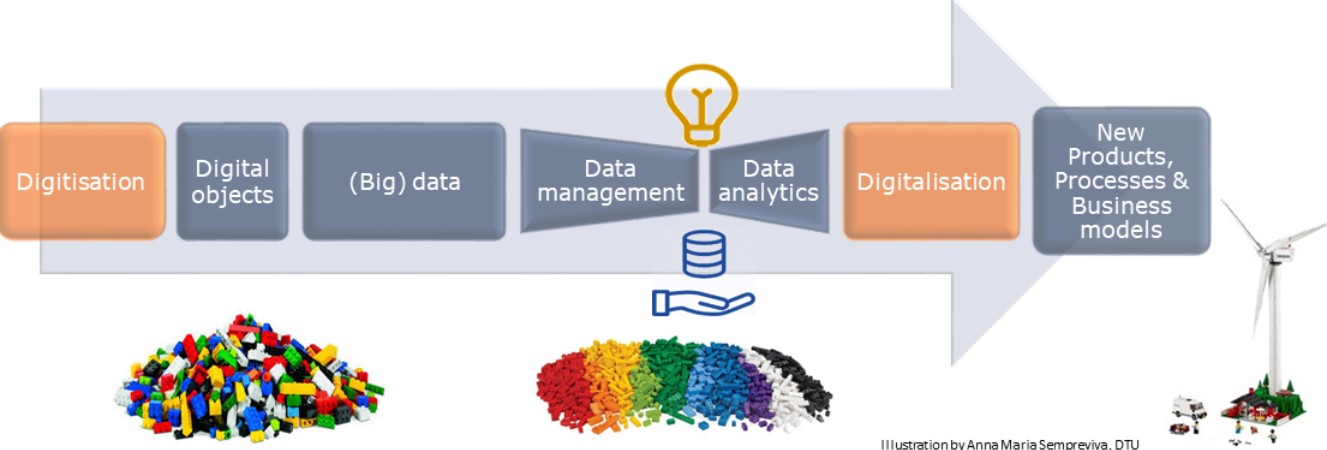

**Figure 6.** Digital transformation is a process that goes from digitisation to digitalisation. After Sempreviva (2020)

## 3.1 Digitisation: converting information from analog to digital

Digitisation is the process of converting information into digital signals (Rachinger et al., 2019), which fuels the digitalisation process. Digitisation can take many forms. One common example is the digitisation of sensor signals from infrastructure throughout the wind energy system. Other examples include construction and maintenance records. And, processes - such as maintenance scheduling - and algorithms - such as energy trading decisions - can also be digitised so that they can be captured in software. This may also mean formalising processes, although it may be more sustainable and flexible to break longer

processes into modules that can then be combined as required. The aim of digitisation is to provide the raw material for later use in digital business, and so it is important to capture as much information as possible in digital formats.

Sensor information from hardware in the wind energy system will be fundamental for releasing the opportunities of digitalisation. Wind turbines and other hardware in the wind energy system are extensively instrumented, and the data are brought together in data systems (often known as Supervisory Control and Data Acquisition systems or SCADA systems). This simpli-

fies the process of data access. However, as we noted in our scenario, turbines are usually only instrumented with the sensors needed to support the manufacturer's and operators' needs, which might be less than are required for some applications. This helps reduce costs of manufacturing and sensor maintenance. Also, although extensive data buses are in place at most wind farms to connect wind turbines and the plant controller, security limitations often mean that it is impossible to connect other sensors to these networks, and that it might only be possible to access sensor data through the turbine's own SCADA system.

The lack of access to turbine data limits the development of software-based third-party solutions, while the lack of data connections means that third-party solution providers often have to develop both the sensor and the data transmission tools, further raising costs. To avoid a situation in which multiple entirely new networks are built on top of each other in future wind farms, it would be helpful for wind farm developers and their engineers to plan in a high-bandwidth open network on new wind farms, with strong security implemented at the hardware level (Ahmed and Kim, 2014).





For all that wind energy uses wind for its fuel, there is a relative lack of information about the wind fields in and around wind farms. This may change through the deployment of remote sensing by lidar, sodar, and radar, or mobile *in situ* sensing by drones (Figure 1). The extra costs of such devices will have to be justified by reduced turbine fatigue, or flexible energy markets where an operator can make an informed cost-benefit decision. As wind lidars have not yet become standard equipment on wind turbines or at wind farms, there is evidently still work to be done to justify their integration.

Digitisation costs money. Therefore, without a clear business case or plan for using digitised data as part of a process of digitalisation, there may be a perception that digitisation is expensive, along with some resistance to starting the process. Although—as in Alice's case—some companies may choose to gamble that later investigators will be able to monetise data, others will require a cost-benefit analysis. While researchers have been able to show potential benefits from digitisation and subsequent digitalisation, for example, by reducing fatigue loads or reducing actuator use, they are often unable to convert

these into meaningful fiscal benefits; in this example, collaboration is required between turbine OEMs and plant operators to develop baseline cost models that could be used for such studies. Other factors can contribute to a reluctance to digitise, for example, the cost of storing data, the difficulty of creating metadata (i.e., information about the data), or the need to develop new processes to deal with sensitive data. However, without digitisation, digitalisation cannot start.

## 3.2   Digital objects

Digitalisation requires that people or machines can find and make sense of the data created through digitisation, and then act upon it. Data alone (i.e., raw binary records) become usable through the addition of human- and machine-readable metadata, that describe and package the data and place them in context, which turns data into discrete "digital objects" (Schwardmann, 2020).

    Digital objects can be created from many different sources. As well as the turbine sensor readings that Alice was using, they

might include research reports, software models or tools, algorithms, and many other things. But these digital objects alone cannot be used effectively as part of a process. Instead, they need to be made findable, accessible, interoperable, and reusable (FAIR; Wilkinson et al., 2016):

- **Findable** means that data can be discovered by people or machines through a search engine. Data can be made findable by identifying the data through the addition of metadata (a list of terms) which follow a defined schema. The metadata

can then be aggregated for a dataset, and users can search through it to find the specific data they require. The process of making data findable adds time and cost to the creation of data, but if data are not findable, they will never have the chance to generate value.

- Once the data has been made findable, it needs to be made **accessible**. This means that the data should be retrievable using secure but open and free protocols, for example through the internet.

- It is rare that one piece of data is sufficient. Therefore, data needs to be **interoperable** so that it can be used in a workflow or applications. In practice, this means the need for machine-readable data and relationships.





**Figure 7.** Combining data from sensors across all parts of the wind energy system will be key to transitioning from today's isolated assets (top) to the interconnected wind plant of the future (bottom). Illustration by Josh Bauer, NREL from NREL/TP-5000-68123 (Dykes et al., 2017). Used with permission.

– To maximise the benefit of sharing data and digitising processes, data should be **reusable** so that they can be applied to different settings. Reusability by anyone besides the original creator requires that data be findable, accessible, and interoperable.

Making data FAIR enables their effective use as part of a process and makes it easier for a digital business to obtain and use the data they require, and is thus an important step in digitalisation. For example, the creation of digital objects makes it feasible to implement large "data lakes" of all kinds of data in multiple formats, instead of rigid warehouses or structured relational databases (Mathis, 2017). In data lakes, digital objects can be added to the lake with appropriate metadata and then retrieved as required. Indexing services offer the potential to create distributed data lakes that cross organisational boundaries,



for example for research data so that users can access data from an ad-hoc mix of sources (e.g. for the New European Wind Atlas Hahmann et al., 2020; Dörenkämper et al., 2020). The use of data lakes reduces the overhead required with maintaining a data structure, such as weather observations organised by year, month, and day. In some cases, the transition away from organised or structured data to data lakes can be challenging. It can be perceived as risky, as unstructured data can become hard for humans to find and relies instead on software to access data. Although this risk can be mitigated through software testing,

user training, and other measures, the perception of this risk is one example of the nontechnical challenges that are faced in digitalisation.

There are a number of other challenges associated with making data FAIR. These include agreeing upon a common metadata schema, deciding on common variable names and data structures, adoption of common metadata schema or data structures, providing incentives for making data accessible (e.g., through marketplace concepts), and developing a common platform to

do this. The European Commission pushed the use of the FAIR principles by researchers through several initiatives; applicants for research funding were partly assessed on their plans to provide open access to data and publications; and indirectly funding the development of sector-specific taxonomies that have made it easier to apply metadata and make data findable. An example is the prototype of the Sharewind metadata registry (Sempreviva et al., 2017) created by the members of the European Energy Research Alliance Joint Programme on Wind Energy (EERA JP Wind Energy) partners in the European Seventh Framework

Programme (FP7) Coordination Action project Integrated Research Programme on Wind Energy (IRPWind). The official adoption of open science for "Horizon Europe" projects in 2018 will also likely lead to more awareness of the FAIR principles. In addition, the United States Department of Energy issued a funding opportunity announcement (FOA) in 2020 for artificial intelligence frameworks (including data) that utilise the FAIR guidelines and principles:

"The DOE SC program in Advanced Scientific Computing Research (ASCR) hereby announces its interest in mak-

ing research data and artificial intelligence (AI) models findable, accessible, interoperable, and reusable (FAIR) to facilitate the development of new AI applications in SC's congressionally authorized mission space, which includes the advancement of AI research and development. In particular, ASCR is interested in supporting FAIR benchmark data for AI; and FAIR frameworks for relating data and AI models." (From FOA DE-FOA-0002306, U.S. Department Of Energy, 2020)

While the FAIR data principles are an enabler of open science – the shift to collaborative scientific development based on transparent and accessible knowledge (Sempreviva et al., 2019) – FAIR data are not necessarily open data. Instead, FAIR data are a prerequisite for openness following the principle that " data should be as open as possible, as closed as necessary" according to the H2020 Program's Guidelines on FAIR Data (European Commission, 2016).

### 3.3 Big data

Digitisation will change the characteristics of the data available to people and organisations. Data volumes increase—hence "big data"—the data are moving or changing at higher velocity, the data are only useful if they can be given value, the data come in many different varieties, and the veracity of the data needs to be checked and ensured. These challenges are known





as the "five Vs" of big data for the volume, velocity, value, variety, and veracity of data (Ishwarappa and Anuradha, 2015).
Practically, data becomes "big" when it cannot be handled using existing approaches. For example, when data volumes exceed

storage capacities on a typical desktop computer, they require specialised storage and analysis tools. As a result, what is
considered big data is not fixed, but changes as storage gets cheaper, computing power increases, or new software is developed.

Although it brings challenges, such a level of data availability also brings advantages, for example the ability to aggregate
information across a fleet of wind turbines, or to look at an entire wind plant project from first idea through to decommissioning.
Furthermore, this ability to see the whole picture allows organisations to identify, explore, and act upon hidden relationships

when appropriate data management and analytics tools are available.

Many tools have already been developed to help users deal with big data. One way to cope with big data is to store it together
with metadata to enable its management. Cloud-based computing allows users to manipulate much larger amounts of data than
can be processed on a desktop computer, while dedicated search tools allow the data to be queried. Similarly, visualisation can
help understand big data (Bush et al., 2017).

Another challenge with big data is the need for organisational IT services that span physical infrastructure and cloud-based
or remote infrastructure. This can bring pricing uncertainty and also increases an organisation's attack surface and can also
require very specialised knowledge. An alternative is to devolve responsibility to individual users, but this can rapidly result in
a patchwork of independent solutions.

Taking cues from the tech sector (e.g., LinkedIn, Google), we expect that metadata will be an increasingly important aspect

of big data analytics. A database noting previously useful connections, verification studies, and framing a general knowledge
map will likely assist in developing next-generation analysis approaches. For example, a knowledge map could assist a large
owner in recognising and connecting maintenance data from a diversity of contractors. This is known as the "open-world"
assumption, where it is not assumed that all data will be acquired in a standard form.

The issues and challenges of big data are generally well-known, and solutions exist. The next step in digitalisation is then to

make data available to the right users to provide insight and apply outcomes. This requires effective data management and data
analytics.

### 3.4 Data management: right data, right user, right time

A digital process requires access to data, such as sensor data, data processing or simulation codes, or other types of digital
information. Data management policies describe how those data are collected, stored, and preserved, as well as who can access

what data. Data management is therefore required to get the right data to the right users within an organisation, and to manage
data sharing across organisations. It is one of the steps for making data FAIR, as discussed in Section 3.2.

Data management also encompasses data discovery and data sharing. Data discovery is the process of finding the appropriate
data in a data storage. Although this can be done by querying the data directly, it is easier to search for data when they are
tagged with the required metadata. Ideally, metadata would follow a standard schema so that all data can be queried at once.

Metadata can be made even more useful if the schema are populated using values from a controlled vocabulary (e.g., a sensor
might measure a physical property that is one of "wind speed", "air temperature", or "air relative humidity"), rather than letting





the user choose freely (Lund and Sempreviva, 2019). This reduces the number of terms that need to be used in searching for data. This approach has been used in the recent IEA Wind Task 43 Standardized Measurement Data Model (Holleran et al., 2022).

These controlled vocabularies can have whatever structure is required. For example, they could just be a list of terms, or could have a hierarchical structure that also includes relations between terms; these are known as taxonomies. They could also include relationships between vocabularies which are known as ontologies. Describing data by tagging it using these ontologies creates so-called semantic data models which further enables data discovery. The IRPWind registry prototype *Sharewind.eu* was one early use of a taxonomy in the wind energy community. This bottom-up effort to compile a metadata catalogue for

wind energy was set up by members of the IRPWind project. *Sharewind.eu* catalogued data using a metadata schema and related taxonomies based on expert elicitation (Sempreviva et al., 2017).

Different domains may have their own ontologies and thus use different approaches to naming a variable depending on historical reasons and the importance of the variable in each field. For example, the wind energy community might describe a physical property measured by a thermometer simply as "air temperature", while the meteorological community might tag the

same data using the class "atmosphere" with a sequence of children (e.g., "dry bulb temperature" and "wet bulb temperature", etc.). Such varied classifications prevent the easy exchange of digital objects across domains. Avoiding this discrepancy across domains is difficult, but it can be mitigated by making equivalences between terms, making the ontologies public and visible, and leveraging existing ontologies rather than developing new ones.

Following data discovery, sharing data is required if the data is to be used in a productive way by the organisation. Sharing

can take many forms, including sharing data within a single team in an organisation to work on a problem together; sharing the data with different teams or units within an organisation to benefit from different knowledge and capabilities of different teams; sharing the data with partner organisations to get the most out of the data; sharing the data with clients; sharing the data with specific researchers within a project to work on R&D projects together; or even sharing the data with the wider community in an open innovation process. Data sharing can be made easier by using common metadata schema across organisations, and by

developing data portal(s) that can query metadata catalogues at multiple organisations.

Willingness to share data is often linked to its perceived commercial importance. Commercially sensitive data might only be shared with a small group and needs to be protected and will almost never be shared outside of an organisation; this is usually known as closed data. Some data might be made available to a small group outside the organisation, or more broadly but under specific licence conditions. These are known as shared data. Other data might be freely accessible to anyone without

restrictions; this is open data.

Sharing data or making it openly available would be a change from today's protected data and would require a new mindset, which could be encouraged by providing examples of the benefits of data sharing and its safety. One potential approach to get the wind energy sector used to sharing is to map and link databases that are distributed within and over different organisations. Each organisation then compiles and exposes a metadata catalogue of the data available for sharing. Then, a web crawler

harvests the metadata and updates a central catalogue that is searchable by the community members. This approach has been proposed by the EERA JPWind Community with the data registry ShareWind. This approach makes data discoverable but not





directly accessible; data are protected by the owners who can then make it available to approved users. In 2021 the EU H2020 Coordination action project FarmConners decided to build on the ShareWind data registry and set up a searchable metadata catalogue called *Share-Wind.com* to make data truly FAIR. *Share-Wind.com* is being populated by different communities
including the New European Wind Atlas NEWA, FarmConners, IEA Wind Task 41, and others. Several other platforms exist to share FAIR data with the wider scientific community, for example *zenodo.org*, and some wind energy-related communities have started to build up data collections on them. The wind energy industry is also starting to make data available; the Danish energy company Oersted has provided access to data from several sites since 2018, and in 2021, SGRE announced a data portal that can be used by industry and researchers. These few initiatives highlight (by comparison) that most energy-sector data is
closed data and cannot be accessed.

A significant challenge to data sharing outside organisations comes from the need to agree on the terms of use for the data. This is usually done through licenses. But there are many potential licenses that can be chosen; as an example, Zenodo allows users to license data under one of 489 pre-existing licenses, or to add their own[4]. The need for data providers to choose a license and for data consumers to understand the license, costs time and requires specialist support. Licensing can therefore
be expensive and slow, and so may act as a barrier to sharing by organisations. Some solutions to this challenge that are being worked on to the knowledge of the authors include:

– Digital Rights Management (DRM) solutions, which aim to control the use, modification, and distribution of copyrighted information.

– Data licensing, whereby asset owners and operators generate extra royalty income at no extra cost by providing data
platform operators the opportunity to monetise the sales and market intelligence value of their data on their behalf, for example.

– Pre-emptive licensing has been used by The Open Energy initiative. Participants in the initiative were required to agree to a common license to be able to participate.

– Data leasing, in which access is granted to data through an API where all the data analysis scripts are monitored.
– Encrypted analysis, in which algorithms are delivered to the data rather than the other way round.

– Federated analytics and differential privacy machine learning techniques, where data is virtually aggregated, allowing organisations to anonymously pool their data.

### 3.5 Data analytics

Data analytics is the processing of data to find insight. It includes inspecting, cleaning, transforming, and modelling data.
Although data analytics have been used in the wind energy industry for some time, the increased amount of data resulting from

---

[4]The complete list of licenses available through Zenodo can be found at https://zenodo.org/api/licenses/?page=1&size=1000.





digitisation, and the increase in computing power (following Moore's Law and through the availability of cloud computing) will give data analytics an opportunity to add tremendous value and even uncover new business models.

With the availability of more data, both historical and real time, models will be able to scale accordingly to be deployable in the real world. Bach-Andersen et al. (2015) highlights this difficulty in wind energy when explaining "A big challenge such

organizations now face is the question of how the massive amount of operational data that are generated by large fleets are effectively managed and how value is gained from the data". This will present a new and unique problem to the wind energy industry regarding how to tailor previous solutions to larger scale data and opportunities to distil information to then be used in real time. One example of this is in the analysis of vibration data for failures (Koukoura et al., 2019), another is condition assessment for wind turbine gearboxes (Zhu et al., 2019), or predicting failures with weather data (Reder et al., 2018).

Data analytics frequently uses models to help understand data. These models can take many forms. For example, they could be models of a physical process such as the weather, or simulations of goods flowing through a manufacturing plant, or a simple statistical model used for measurement instrument calibration. These models help identify deviations from expected performance given known inputs or forecast future states. The availability of increased computational power allows the use of more complex models, but such models are expensive to operate and maintain and require specialist skills. In contrast,

Rezaei et al. (2015) note that "... reduced order models (ROM) are more desirable due to less computational cost and enough accuracy". ROM are especially important for real-time or near-real-time applications such as energy forecasting and grid integration. These applications highlight the need for increased model accuracy and speed improvements, not only additional computational resources. Data-driven and physics-informed models can fill this critical need for faster, less resource-intensive ROMs.

There is also an additional opportunity with federated learning and privacy-preserving machine learning to collaborate across organisations and share results as well as models. Lameh et al. (2020) explain the introduction of federated learning by McMahan et al. (2017) as "a decentralized ML approach suitable for edge computing", with an ability to respond to processing needs. This will bring the ability to advance the wind energy field without the concerns regarding data privacy. Federated learning and privacy-preserving machine learning can enable the deployment and training of a global model system where

multiple independent organisations contribute their data (for example, sensor data from a specific wind turbine type) and all users benefit from more accurate models based on more training data, but contributors only ever have access to their own data.

These new opportunities will bring new challenges. With the opportunity to collaborate across the industry and share more information, questions will arise regarding data ownership and control of models.

### 3.6   Digitalisation

As noted in the introduction, we define digitalisation as the organisational and industry-wide use of data and digital technologies to improve efficiency, create insights, and develop products and services. Digital technologies typically span from sensing to data management to data analytics to artificial intelligence and automation. In short, it is doing new and value-driven things with digital tools.





The previous sections discussed the combination of technical and cultural measures that are needed to bring data to users
to assist informed decision-making. They can form part of a workflow that brings data from sensors to users, allows them to
make decisions, and then passes information about that decision to other users (as for smartphones) or back to hardware-level
actuators (as in the Internet of Things). Although many individual workflows might be digitised and rely on digital objects and
data management to get data to users, and then make decisions based on data analytics, this does not mean that the full benefit
of digitalisation has been realised, as digitalisation also benefits from network effects. As more organisations undergo a digital
transformation, they will be better able to interact with other digitalised organisations, forming a virtuous upward spiral.

One challenge with digitalisation is that there is an adoption process. Potential users of digitalised workflows span the
usual range from innovators, early adopters, and mass-market through laggards; initial successes by innovators lead to more
awareness and increasing adoption, before an innovation becomes "normal" and is used by the mass market. The adoption
process therefore needs to happen both within organisations and in the wider wind energy industry.

The challenges within organisations relate both to people and culture. We have already noted that adoption of digitalisation
will require skills such as computer programming, but it will also require that these be better integrated into existing training
for science, technology, engineering and mathematics (STEM) subjects and professional trades. Users will need to be given
the opportunity to apply digitalisation—requiring that their organisations support innovation—and will need to be rewarded
for their efforts. Furthermore, organisations need to understand the potential value of their data (in all its forms) and develop
processes that simplify data-led collaboration. In some cases, these changes may be difficult or disruptive to implement within
existing structures, and it may make sense to create in-house incubators, accelerators, or skunkworks to enable innovation.

The wind energy industry needs to explore ways to encourage digitalisation. One way to do this may be to demonstrate
the benefits of digitalisation. This could include nationally funded pilot or lighthouse projects but could also include market-
places to connect innovators and early adopters in rewarding partnerships; one option here may be to (further) open existing
marketplaces such as day-ahead energy trading or contract-for-differences schemes and give digital solutions a low-risk envi-
ronment for testing, where failure or malfunctions have limited or no impact on the rest of the system. These environments are
often known as sandboxes. As well as digital sandboxes, hardware testbeds are also required to allow solutions to be trialled
in realistic settings where risks can be managed. This could include wind turbines from a few hundred kilowatts through to
multimegawatt machines (allowing solutions to be transitioned into products) or even multiple wind turbines within operating
wind plants. Importantly, these sandboxes and testbeds would allow benchmarking against traditional approaches, showing the
benefits of digitalisation. These initiatives would also have intangible benefits such as increased awareness, developing trust,
and creating networks of service providers.

## 3.7 New products, processes, and businesses

The culmination of digitalisation is new modes of business, new ways of doing things, and new means of interacting with
people and machines (Ignat, 2017). It is more than simply combining all these features into the latest gadgets but is about
effective ways of organising human efforts to make the most out of these innovations. Taken all together, these new models





lead to "enabling various new forms of cooperation between companies and leading to new product and service offerings as well as new forms of company relationships with customers and employees" (Rachinger et al., 2019).

New technologies can bring about new businesses, the emergence of failure prediction models, and benefits from predicting failures, but coupled with the open shared data, a few companies can now specialise in failure prediction, providing the service to others and bringing huge economies of scale to the whole industry. Not only is a new company created, but all the clients of the new company receive its benefits. This is particularly the case if digital processes can be turned into software and sold repeatedly (Berkhout et al., 2020).

Properly resourced and directed research institutions and industry organisations are the start to spurring research connected to real problems in the field. The research funding needs to be willing to fund efforts further along the innovation pipeline to help bridge the so-called "valley of death" that separates research from commercially-relevant products. Industry needs to be properly developed enough to be willing to take risks in reaching backwards across the valley of death. Industry groups need to be well developed enough to guide all actors and instil enough trust and industry guidance to know that their investments will work out and all actors agree on base standards and overall direction.

The organisations need to be designed to be willing to make use of these innovations. Chesbrough (2010) discusses how without the capability to innovate their business models, companies will have no means of commercialising their ideas and technology. Organisations must have communication to spread ideas across the company, culture which allows people to speak up, leadership that fosters free thought, and autonomy that allows every employee to act.

Leaders should describe all these efforts in a digitalisation strategy that sets out "a commitment to a set of coherent, mutually
reinforcing policies or behaviors aimed at achieving a specific competitive goal" (Pisano, 2015). These organisational principles are true whether the organisational unit is a company, research institute, or government.

Moving beyond simply implementing new technologies into creating entirely new markets requires an ecosystem of new ideas to develop. For example, without a wide combination of technologies, drones for offshore wind would not be possible. This level of innovation requires coordination across organisations. Drones require battery tech, electronic controls, cameras
and sensors, data connectivity, and AI control. The potential market of drones for offshore wind energy applications will bring about numerous companies for the support of all these factors. This is encapsulated by Christian Burmeister and Piller (2016), who define business models as "as a management hypothesis about what customers want, how they want it, and how the enterprise can organize itself to best meet these needs, get paid for doing so, and make a profit". The integration of all these aspects, technology, product, customer, and employees, play a part in higher-level digitalisation efforts that provide real value.
How can this vision be implemented in reality? Some of the many relevant initiatives in the wind energy community that aim to bring together all of these needs by sharing data and knowledge to mutually benefit the entire wind energy industry are listed in Table 3.

## 3.8 Summary

Based on the previous sections, the key aspects of the digital transformation most relevant for the wind energy industry can be
summarised as follows:



**Table 3.** Digitalisation initiatives in the wind energy industry. The examples are illustrative only and not meant as an endorsement; many other alternatives exist.

| Type | Activity | Examples |
|---|---|---|
| New products | Data marketplaces | Greenbyte marketplace for wind data |
| | Knowledge sharing | IntelStor Market Intelligence Ecosystem |
| | Data discovery and sharing | Share-wind.eu metadata catalogue, WP3 Benchmark (Fields et al., 2021), US DOE Data Archive & Portal (Macduff and Sivaraman, 2017) |
| | Data services from turbine manufacturers | GE Digital Wind Farm Services and Solutions |
| | Newly enabled products | Falco Drone Technologies used for offshore turbine inspection (Buljan, 2020) |
| New processes | Comparison and benchmarking activities | IEA Wind Task 31 (Wakebench; Rodrigo et al., 2014; Moriarty et al., 2014), IEA Wind Task 30 (OC6) floating wind turbine benchmark exercise (Robertson et al., 2020), CREYAP: Comparison of Resource and Energy Yield Assessment Procedures (Anderson and Mortensen, 2013) |
| | Collaboration platforms | *WeDoWind* data sharing and collaboration platform |
| | Open-source tools | Brightwind, The OpenOA codebase for operational analysis of wind farms (Perr-Sauer et al., 2021), Data Science for Wind Energy R Library (Kumar et al., 2021) |
| | Open Data Standards | IEA Wind Task 43 WRA Data Model (Holleran et al., 2022), ENTR Alliance, OSDU |
| New businesses | Data-specific companies | Atrevida Science, WindESCO, Clir, PowerFactors, and i4see |
| | Companies outside of wind energy applying solutions to wind energy | Uptake applying failure prediction models to wind energy. |

- **From digitisation to digitalisation:** The measurement and storage of data that can be used to make business decisions can be time-consuming and expensive, and new methods and systems are required.

- **Digital objects** need to be FAIR so that they can be used effectively in the digital transformation.

- **Big data:** Handling the large amount of data involved in digitalisation requires both semantic and technical solutions. Semantic solutions includes standard metadata and related ontologies, while technical solutions include either new efficient storage systems or new strategies such as linking data distributed across organisations.

- **Data management:** A data management policy concerning the way data are collected, stored, and preserved is crucial for getting the right data to the right users within an organisation, and for data-sharing amongst organisations.

- **Data analytics:** New innovations such as federated learning and privacy-preserving machine learning are required to reduce the computational power required for data analytics applied to big data.





- **Digitalisation:** The new processes and ways of working required for successful implementation of the digital transformation need to be fully adopted by people within organisations.

- **New products, processes, and businesses:** The successful transformation of new findings based on data analytics into new products, processes and business can only happen if research institutes, funding bodies and industry partners actively come together to bridge the valley of death. This could be achieved through new digitalisation strategies.

## 4 Lessons Learned from Other Networked Industries

Wind energy is characterised by a large amount of dispersed infrastructure that generates lots of data, a flexible network between that infrastructure and other systems, and many different stakeholders. Those stakeholders need access to widely varying data and have different abilities to act within or upon the system, or with each other. Together the wind turbines, stakeholders, and service providers form a digital ecosystem.

These characteristics are also found in other sectors. For example, the Internet of Things (IoT) and smartphones also combine networked hardware, large numbers of users, and large volumes of data. In this section, we examine these sectors and summarise the lessons learned from their specific challenges that could be transferable to the wind energy industry.

### 4.1 The Internet of Things

The Internet of Things is a web of a multitude of physical objects—things—that are connected to each other, other systems, and users through the internet. It is gradually becoming a reality as more goods are equipped with sensors during manufacturing and can be connected to ubiquitous internet connections. It leverages cheap sensors and easy access to the internet to provide regular data transfer.

The IoT can be thought of as having several discrete layers that facilitate the movement of data between hardware and users, and informed interaction with the hardware (e.g., Sethi and Sarangi, 2017). These include:

1. The perception layer, where data is obtained by sensors and actuators can act upon infrastructure.

2. The transport layer, where data is collected from sensors and made accessible for processing. Services in this layer also provide secure access to the sensors.

3. The processing layer, where sensor data are stored, aggregated, and interpreted.

4. The application layer, where data is visible to users through specific interfaces, for example for building management, health, or other applications.

5. The business layer, where decisions are made with relation to a person's motives or business's goals.

This layer approach allows data to be transferred up to users and back to actuators through specialist service providers. A user can then use vertically integrated solutions or combine different services (potentially with the help of an integrator) to get the solution that they need.





The IoT enables many different applications, including vehicle fleet monitoring, automated inventory management, and monitoring national infrastructure. It also has applications for consumers, for example for access passes at entertainment parks (e.g., Disney's "Magic Band" Marr, 2016) or for order-on-press buttons (e.g., Amazon's "Dash" button). And, the internet of things is already enabling wind turbine condition monitoring (Coronado and Fischer, 2015).

Adoption of the IoT has not been uniform. The major challenges to adoption reported in a 2016 survey included (in descending order of frequency) privacy and security concerns, the cost, lack of knowledge about solutions, inadequate infrastructure, lack of standards, interoperability concerns, uncertainty that it will deliver the promised benefits, poorly defined workflows, and immature technology (Buntz, 2016). The same challenges should be expected for the digitalisation of wind energy, which is in many ways just an application of IoT approaches.

## 4.2 The smartphone hardware and software ecosystem

Smartphones are small, portable computers with an internet connection. They run specialised applications or "apps" to carry out a wide range of tasks, from telephony to web browsing, shopping, participating in social networks, and for many other purposes.

The apps run on the smartphone's operating system. The apps leverage other apps or service providers to identify users,
manage their rights to interact with the app, and charge them for services. From the user's perspective, this is mostly seamless, and improvements can be made by modifying software, so long as the hardware is flexible enough. Apps allow consumers to pay for what they consider important and customise their smartphones to meet their needs. Consumers have demonstrated their willingness to pay for apps: in 2020 582 billion USD of revenue was generated through apps alone.[5]

In the early days of smartphones, vendors often created "walled gardens" in that each vendor had their own operating system,
effectively locking customers and software developers into that platform (Eisenmann et al., 2008). Today, smartphone vendors instead use several common operating systems, allowing software developers to target the operating system and not the vendor, and enabling customers to keep their apps and data, even if they change vendor. Those smartphone businesses that are vertically integrated in that they develop and sell smartphone hardware, operating systems, and apps, almost all allow third-party services to be run on the handset or operating systems. This change was in response to consumers requiring interoperability across
hardware, and the growth of providers who provide apps for multiple operating systems. However, it has also raised concerns about the ability of handset manufacturers to access data stored on those devices.

The smartphone market includes traditional business-to-consumer (B2C) services such as selling the phone hardware, selling one-time services (e.g., apps), subscription services, as well as disruptive innovation and experimentation (e.g., Uber and Instagram). It also includes large business-to-business (B2B) services, for example selling network infrastructure, hardware
components, software, and back-office services. In 2020, around 1.4 billion handsets were sold for around 400 billion USD,[6] while the telecom network equipment was worth another 500 billion USD in 2020.[7] Each of these related markets grows at between 2% and 10% per year as technologies are updated, new businesses are launched, and customer expectations change.

---

[5]https://www.statista.com/statistics/269025/worldwide-mobile-app-revenue-forecast/
[6]https://www.statista.com/topics/840/smartphones/
[7]https://www.verifiedmarketresearch.com/product/telecom-equipment-market/





### 4.3 Lessons for the future wind energy digital ecosystem

The relevant lessons learned from the examples above can be summarised as follows:

1. The digital future will be dynamic, with an unpredictable mixture of actors and technologies.

2. There will be very few, if any, vertical solutions that directly connect the user with the hardware. Instead, most interactions will be between one layer and the next.

3. Standards about interaction between and within different layers will be required to avoid market fragmentation.

4. Specialist service providers will develop to support activities. These may be within a single layer, or link one layer to the
next.

5. Services such as user authentication, permissions management, intrusion detection will be essential for safe and reliable operation of the system.

6. Innovation will happen in unexpected places and will take unexpected forms.

7. It will involve a competitive, but collaborative, community.

These lessons learned should also be considered by the wind energy industry as it undergoes its digital transformation.

## 5 The Grand Challenges

In this section, we first describe how we identified the Grand Challenges, and then we introduce them.

### 5.1 Methodology for identifying the Grand Challenges

The Grand Challenges for the digitalisation of wind energy were identified by:

1. Consideration of the conclusions from the previous sections of this paper.

2. Further stakeholder data collection, including data sharing interviews and around diversity in wind energy.

3. Combination of findings from steps (1) and (2).

These steps are described further, below.

#### 5.1.1 Consideration of the conclusions from the previous sections of this paper

The conclusions from the previous sections of this paper that are relevant for identifying the Grand Challenges include:





– **Data:** A data management policy concerning the way data are collected and stored and preserved is crucial for getting the right data to the right users within an organisation and data sharing amongst organisations. The measurement and storage of data that can be used to make business decisions can be time-consuming and expensive, and new methods and systems are required. To utilise data effectively for the digital transformation, it needs to be FAIR. The large amount of data involved in digitalisation means that new, efficient storage systems and agreed-upon metadata standards are required. New innovations such as federated learning and privacy-preserving machine learning are required to reduce the required computational power of data analytics applied to big data. Services such as user authentication, permissions management, intrusion detection will be essential for safe and reliable operation of the system.

– **Culture:** The digital future will be dynamic, with an unpredictable mixture of actors and technologies. Innovation will happen in unexpected places and will take unexpected forms. The new processes and ways of working required for successful implementation of the digital transformation need to be fully adopted by people within organisations.

– **Coopetition**: The successful transformation of new findings based on data analytics into new products, processes, and business can only happen if research institutes, funding bodies, and industry partners actively come together to bridge the commercialisation valley of death. This could be achieved through new digitalisation strategies. There will be very few, if any, vertical solutions that directly connect the user with the hardware. Instead, most interactions will be between one layer and the next. Standards about interaction between and within different layers will be required to avoid market fragmentation. This will require a competitive, but collaborative, community.

Considering the conclusions from the previous sections of this paper allows us to make two further observations relevant for identifying the Grand Challenges. The first is that, despite the issues identified in the stakeholder interviews and in this paper, the digitalisation of wind energy is already progressing. Some organisations have made progress in adopting it for parts or all of their businesses. Evidently, the challenges identified by the survey and expert elicitation can be mitigated, and the technical solutions required for digitalisation must exist. Therefore, "the future is already here — it's just not very evenly distributed" (attributed to William Gibson, possibly apocryphal). The second observation is that, because of its cyclical and transitional nature, the process of digitalisation will never be complete.

### 5.1.2 Data collection activity A: data sharing interviews

The data collection activity focused on the specific barriers seen by the wind energy sector in the actual implementation of digitalisation. For this, a total of 30 members of the global wind energy sector were asked to describe their main barriers to data sharing in individual interviews lasting approximately one hour. This included seven owner/operators, seven researchers, and sixteen technology providers from the wind energy industry. This specific question was asked because it has already been established that the topic of data sharing and reusing is one of the central challenges in the implementation of digitalisation in wind energy (Figure 4). The survey was done as part of the activities of IEA Wind Task 43 Technical Area 2. The results are summarised in Table 4. This shows the five barriers with the highest frequencies of occurrence for each interviewee type.





**Table 4.** The top five barriers to data sharing. These barriers were identified from the second interview series.

| Rank | Owner/operators | Academia | Technology providers |
|---|---|---|---|
| 1 | Getting all the data in one spot | Lack of public data | Data quality (completeness, validity, etc.) |
| 2 | IT issues: servers, etc. | No standard format for analysing and processing data | Different format and structure of data |
| 3 | Cleaning/filtering raw data (different time scales and resolutions, different formats) | Poor data quality | Data filtering for analyses |
| 4 | Refining and processing data ready for machine learning model (80% of time) | Lack of willingness to share data, especially higher resolution | Data collection; different devices need to be programmed differently |
| 5 | Interfaces to collecting data reliably | Lack of change logs | Time for downloading, cleaning, and training data |

Some of the barriers are quite similar between interviewee types, in particular the topics of (a) standard data format/structure, (b) data availability and quality, and (c) data processing/preparation time.

Other barriers that came up frequently but did not make the top five include:

- Unclear intentions of OEMs/unannounced changes to controllers.

- State-of-the-art in data science not clear, no place where work is well-summarised.

- Differences between different analysts. What really works and what's really applicable?

- Data security and privacy

- Finding the right people who can do data analytics and have the required domain knowledge.

It should be noted that these findings generally agree well with the literature survey in Section 1. As well as this, it is interesting that most of these barriers are related to company culture, people skills, and communication between different people. These aspects should therefore be included in any discussion about the digitalisation of wind energy, and this is the topic of the second interview series described below.

### 5.1.3 Data collection activity B: diversity in wind energy

We have shown that some of the key barriers to the successful implementation of digitalisation in the wind energy industry are related to company culture, people skills, and communication between different people. This raises the topics of Diversity, Equity, and Inclusion (DEI). Diversity is any characteristic that can be used to differentiate groups and people from one another, and thus DEI is about respecting and valuing the aspects that make people different, for example in terms of age, gender, ethnicity, religion, disability, sexual orientation, education, and national origin. DEI is an increasingly important topic especially in technology- and innovation-based industries.





To examine this topic in more detail and in the context of the wind energy industry, a second data collection activity was carried out by the Diversity Committee of the European Academy of Wind Energy (EAWE). This data collection was focused around two main events: a DEI panel discussion at the TORQUE 2020 conference and a DEI workshop series.

The TORQUE 2020 panel discussion was postponed due to COVID-19 and actually took place in June 2021. A literature survey carried out in advance of the event showed that technical workforces in general do not represent the population's diversity in race, gender and sexual orientation. For example, women represent only 21% of the global wind energy industry's workforce and only 8% of its senior management (Ferroukhi et al., 2020). This is known as "under-representation". It is a problem because the literature survey also found that diverse teams are more productive (for example, gender diverse companies

are 15% more likely to outperform their respective national industry medians) and that diversity is good for business (as diverse companies are 35% likelier to financially outperform the industry medians). The panel discussion, and audience contributions from more than 50 members of the wind energy research community during the event, revealed that the community generally sees and understands the benefits of increasing diversity in wind energy, but many people do not know how to contribute. The expert panellists made some suggestions about how to contribute, including sharing facts and stories with each other, noticing

and correcting one's own unconscious biases, and ensuring diverse recruiting teams.

Around 30 wind energy professionals took part in the workshop series. As in the panel discussion described above, it was found that many people simply do not know how to contribute to improving diversity. The main DEI challenges of the wind energy community were found to be (1) attracting, (2) recruiting and (3) keeping a diverse range of people as well as (4) ensuring a diverse range of people in higher management. Possible solutions to these challenges suggested by the community included

organising specific diversity events at conferences and trade fairs, providing a platform for sharing experiences, developing recruiting guidelines, providing diversity training / education for team leaders, providing mentoring schemes, networks and role model events and providing a framework for proactive hires, exchange programmes, female-only positions, and others.

In summary, the main findings of this data collection activity were that (a) diversity has to be improved in the wind energy industry in order to successfully exploit the advantages of digitalisation, and that (b) concrete steps are already being undertaken

by the wind energy industry in order to achieve this goal.

As far as the authors are aware, the specific topic of diversity in wind energy digitalisation has not yet been studied. However, discrimination of under-represented groups caused by digitalisation has been well documented - for example, digital technologies are trained on male-biased data (Feldman and Peake, 2021) or existing recruiting patterns that support homogeneous workforces. These technologies then recreate and reinforce these patterns. This is known as the "digital divide" (Sorgner et al.,

2017). This topic should be further investigated in connection to wind energy.

### 5.1.4   Combination of findings

Based on the observations and interview results presented above, we therefore propose three Grand Challenges for digitalisation that are joined in a cycle that combines the process of technology transitions (as discussed in Section 6) and technology adoption (Venkatesh and Davis, 1996). In this cycle, data are made available, then used as the basis for innovation, and the innovations

are brought to market. The Grand Challenges associated with each of these steps are the need to:



1. Create reusable data frameworks

2. Connect people to data to foster innovation

3. Enable collaboration and competition between organisations.

These Grand Challenges are described in more detail in the following section.

## 745  **5.2   The Grand Challenges**

### 5.2.1   Data: Creating FAIR data frameworks

Digitalisation is, at its root, the ability to act upon data. An example of this is a decision making processes that brings together many different data sources, analyses them, and takes decisions around that data. Because the same data might be used by many different actors and for many different purposes, the data need to be FAIR.

The interviews presented in Section 5.1 confirmed our experience that much of the data generated in the wind energy sector today are not reusable because they are not documented by metadata and thus remain isolated or siloed within organisations or data stores, simply not findable. In the initial interview round, it was noted that one of the main barriers for a successful digital transformation is the "lack of data standards and interoperability". This is a well-known problem, and policy leadership by governments, public bodies, and organisations is helping to mitigate this (e.g., European Commission, 2018). The growing

requirement by funding agencies for data management plans for research projects is helping to train the next generation of wind energy sector researchers to think about data. Reporting requirements also help generate the data that are needed, for example from the U.S. Energy Information Administration about plant construction and performance (the IEA 860 report) or the North American Electric Reliability Corporation (NERC) reports about generator availability (the NERC GADS reports).

    As noted in the interviews, an opportunity for overcoming the challenge of the findability of data is to implement "top-down

policies such as FAIR data requirements for scientific journals and research projects". As introduced in section 4.2, this implies the need for agreed standard metadata schema and related ontologies to help navigate the wealth of data. There have been some efforts to produce these standard metadata schema and related ontologies e.g., by the IRPWIND project and through activities in IEA Wind Tasks 32, 37, and 43. However, making them converge, bringing them to a maturity level and spurring community adoption needs a concerted effort and sector-level collaboration. This is one example of how reusing data can be considered a

grand challenge.

    Reusability also requires ways to get data to the right users at the right time. Once data has been found, it needs to be made accessible to users. This applies equally within organisations and outside of organisations. While internal data access is an organisation's own challenge to deal with, the wind energy industry needs to improve access to data at the sector level, and the interviews revealed that especially industrial stakeholders saw an opportunity in "data markets allowing access to data, which

even can be paid for". As noted earlier, this access to data can often be essential for innovations (Sovacool and Enevoldsen, 2015).



Although they are required to incentivise data sharing, marketplaces for digital services are rare. Some marketplaces exist for data, such as the U.S. Atmosphere to Electrons (A2e) data archive and portal (Macduff and Sivaraman, 2017), while the prototype of the *sharewind.eu* metadata registry from EERA JP Wind launched in 2017 has been upgraded to a FAIR searchable
data catalogue called *share-wind.com* by the H2020 Coordination Action project *FarmConners* and is being populated by different projects. These markets are focused on discovering existing data, rather than buying or selling data. First steps are being taken to make high-frequency real-time data available, which will be essential for more flexible operation of wind turbines and wind plants. True markets for digital services (i.e., where services and data can be bought and sold) are needed so that innovators and early adopters can try them out and develop processes that the majority can then benefit from. Like other
new markets, these will probably be small, illiquid, and unprofitable at first and need some time to generate enough users to be effective. Therefore, national or sector funding may be required to support the initial growth of data and service markets.

It is our opinion that the difficulty of making data reusable across organisations, and the need to collaborate to provide the framework to do so, and the potentially huge reward as a result, makes reusing data into a Grand Challenge for the wind energy industry.

**5.2.2   Culture: Connecting people and data to foster innovation**

It has become clear in this paper that reusing data alone will not allow the full potential of the digital transformation to be exploited in the wind energy industry. As expressed in our qualitative data collection, digitalisation and digital R&D succeeds when connecting data with models with people. This can be done, for example, by developing and maintaining new types of organisational cultures, by combining staff skills and training needs in new ways, enhancing communication skills, working
together on new types of innovation and change processes, and by increasing diversity. However, a frequent point mentioned in the interviews was that "organisational change and change management are the biggest challenges... of digital transformations".

Digitalisation is also a change to established ways of working. One interviewee's comment captured the consensus from many others by noting that "researchers require more digital skills, and furthermore a stronger collaboration with software
developers". It can require new skills from individuals (e.g., computer programming); new ways of thinking (modular and collaborative, as opposed to siloed); and new infrastructure (cloud computing and distributed teams, rather than centralised services). It also requires organisations to embrace open innovation styles such as co-developing new digital products or reconfiguring existing processes to enable new business ideas. And organisations need to bring these innovations to market, while the market needs to be willing to accept new products or services.
The interviews described in Section 5.1 confirmed that some of the key barriers to adopting these new ways of working are actually part of the organisation's own culture, and furthermore in the strategy and perception of other actors. Overall, and regardless of the respondents' background, open innovation and open-source paradigms were determined to be key levers for success with the digital transition.

Organisations that can change their culture become well-placed to benefit from digitalisation, while organisations that are
less able to change risk losing innovative staff to other organisations that can support them. Such support can take many and




varied forms that aim to align internal creativity with the organisation's mission, leading to new products and services (Isaacs and Ancona, 2019). There are many well-established ways to train staff in new skills, change existing corporate culture, or establish new corporate cultures. Therefore, we do not see the process of corporate change *per se* as a Grand Challenge.

Similarly, there are many well-established ways for governments to reduce the risk of innovation. Governments can help to
reduce financial risk by reducing the cost of innovation through grant funding, tax credits, and many other means. Governments and regulators can also help open the window of opportunity by establishing the right market conditions and by providing safe spaces to trial new technologies, such as specially developed infrastructure.

Instead, it is important to recognise that digitalisation is different from other innovation and change processes because it relies on bringing data and people together to trigger and support innovation. Therefore, organisations need to provide the
internal processes for people to access and explore data, mechanisms for these innovations to be identified and nurtured, and spaces to try out their ideas in a business context. Governments and regulators can also support this by creating sandboxes— real or virtual infrastructure—that can be used for pilot projects, and as expressed by a respondent, "understanding that this data revolution brings the energy industry to an inflection point" where regulatory support is required. These are often national or flagship facilities such as NREL's Energy Systems Integration Facility (ESIF) but can also be regional infrastructure.

In the surveys carried out as part of this paper, a common desire among the respondents, regardless of job title, company type, and geographical location, was to enhance the opportunities for collaboration amongst the diverse actors in the energy industry, and beyond. And, improving diversity, equality, and inclusion in the wind energy sector may increase the effectiveness of that collaboration. Long-term initiatives to increase gender equality and increase the workforce diversity are therefore essential to the future success of digitalisation.

We consider the need to connect people and data to foster innovation as a sector-wide Grand Challenge, as it impacts nearly all organisations in the wind energy sector. It requires organisations to offer new approaches to doing business, their customers and partners to request and support these innovations, and government support to be successful. Establishing innovation cultures inside organisations therefore needs collaboration between many stakeholders on a wide range of initiatives, making this a sector-wide challenge.

The result of overcoming this Grand Challenge will be organisations that have innovation cultures (either in parts or as a whole) and can thus respond to the changing landscape around them. This will be an essential characteristic both to drive digitalisation and to benefit from it.

### 5.2.3 Coopetition: Enabling collaboration and competition between organisations

Addressing the Grand Challenges of reusing data and creating innovation cultures within organisations provides the basis for
digitalisation within organisations or narrow sectors. However, one of the biggest potentials of digitalisation is that it allows artificial barriers to be broken down.

Our final grand challenge is therefore to enable cooperation, collaboration, and competition between organisations. This means working together to create marketplaces or business opportunities that would not otherwise exist and that are mutually beneficial. This has happened in other businesses; possibly the best-known example of this is the Bluetooth standard for short-





range wireless data exchange, which grew out of collaboration between device manufacturers in the 1990s. This process is sometimes known as coopetition or co-creation and is the antithesis of vertical integration in which internal R&D leads to internal development, production, and distribution.

There are several steps to be taken before this can occur across the wind energy sector, which we define based on our experience combined with the results of the interviews described in this paper. The first is to simplify the process of coopera-
tion across organisations by streamlining nondisclosure agreements, licensing, and other nontechnical barriers to cooperation. Together with making data reusable, this will mean that the barriers to cooperation are reduced, and is more likely to happen within small, trusted groups initially (e.g., the UK Open Energy initiative). This may turn into a "race to the bottom," as streamlined organisations are easier to collaborate with, and once one business starts the process, others must follow or risk being side-lined.

The next step is to set up liquid, easily accessible markets where innovative, collaborative organisations can meet customers. Such marketplaces will likely make extensive use of metadata, sector-level taxonomies, standardised virtual environments, and open-source data libraries to make their services easily findable, interoperable, and reproducible. Data itself is likely to become a more widely traded commodity. There will be a need for data providers, marketplaces, and data users to implement processes to keep data up to date and maintain interoperability. These processes could leverage approaches used in the software commu-
nity for open-source software versioning, distribution, and compatibility monitoring (e.g., GitHub.com, or PyPI). Otherwise, there is a risk of data becoming unusable, irrelevant, or even misleading over time.

The final aspect of this Grand Challenge is the need to help all stakeholders understand the potential for digitalisation. This could take many forms, from clearly communicating the results of research projects to benchmarking new tools against established ways of doing things, or even by businesses sharing the need for certain new tools. All of these factors together
will help generate demand for the new businesses arising from digitalisation.

Together, simplified cooperation across organisational boundaries and liquid markets makes the collaborative promise of digitalisation a reality. However, these are not the final step in digitalisation; making progress on any of these three Grand Challenges creates the potential for more innovation, new businesses, and new opportunities for coopetition. Therefore, we consider the Grand Challenges to be cyclical, rather than an end-to-end process.
These Grand Challenges must be continually overcome so that the wind energy industry can reap the benefits of digitalisation, such as reduced costs, improved performance, increased safety, and new business models.

## 6 Conclusions

In this paper, we have defined three Grand Challenges in the digitalisation of wind energy. This was done by first exploring digitalisation as a key pathway for the wind industry to be a cornerstone of a decarbonised, low-cost energy future. Ongoing
digitalisation efforts across the wind energy industry were used as examples to identify some of the opportunities and challenges that the industry faces. They also show that the digital wind plants of the future will utilise a host of technologies such as sensors, data, analytics, digital twins, and even automation. These technologies will improve safety, efficiency, and reliability,





all while reducing costs. The vision of the future in this paper is further informed by collaborative research and expert elicitation across many sectors of the industry. This research has helped illuminate the current state of the industry and critical barriers to overcome. Next, we put forward a more detailed and consensus definition of digitalisation to help readers understand the digital transformation and the key concepts underpinning it, as well as to help us define the Grand Challenges in the digitalisation of wind energy. We also looked at examples from other industries that can help us accelerate through the learning curve. We find that walled gardens and vendor lock-in prevent innovation ecosystems. However, ecosystems that are open enough to have common data formats and interoperability can lead to new business models, rapid adoption, and best-in-class offerings.

Digitalisation of the wind energy sector offers increased reliability, cost savings, new business models, and cost-effective integration of wind energy as an energy source. But it brings three Grand Challenges that need to be solved. These are:

1. Data: Creating reusable data frameworks

2. Culture: Connecting people and data to foster innovation

3. Coopetition: Enabling collaboration and competition between organisations.

Addressing the first two Grand Challenges for the digitalisation of wind energy provides the right conditions for sector-wide adoption. Then, the third Grand Challenge brings together actors to make progress. Addressing these Grand Challenges creates a virtuous circle that would lead to increasing digitalisation and increasing benefits to the entire wind energy industry.

Solving these Grand Challenges will require contributions from a variety of stakeholders. To that end, we highlight recommendations for individuals and organisations in Table 5. These recommendations were informed by the literature survey and expert interviews presented in this paper.

These recommendations for ways to address the Grand Challenges in the digitalisation of wind energy provide a framework for a more efficient, inclusive, and innovative wind energy industry. They are only the beginning of a dynamic process that will position wind energy as an essential part of a global clean energy future.





**Table 5.** Recommendations for how different organisations - businesses, research organisations, and funding agencies - could address the Grand Challenges in the digitalisation of the wind energy industry by setting strategic goals and through specific implementation actions.

| Grand Challenge | Strategic Goals | Implementation Actions |
|---|---|---|
| Data | – Organisations should implement tools and frameworks to maximise the value of their data | – Funding agencies should require that all data collected and produced be FAIR<br>– Scientific publishers should require FAIR digital objects (data, code, etc.)<br>– Organisations should participate in sector-level collaboration to define and adopt universal data standards and schema<br>– Funding organisations should support the creation of infrastructure for data and digital service marketplaces built on FAIR practices<br>– Businesses and research organisations should provide ways for internal and external stakeholders to access and explore data efficiently<br>– Organisations should ensure that data systems follow cyber-security best practices |
| Culture | – Digitalisation should be a common thread throughout an organisation's activities. It should be supported by recruitment, (re)training, removing barriers to adoption, and staff evaluation<br>– Organisations should align digital initiatives with their mission, leading to new products and services<br>– Digital initiatives should be people-focused and prioritise stakeholder buy-in as a precondition for success<br>– Businesses should create cultures that allow new types of innovation and change processes to be tried and tested in safe environments<br>– All stakeholders should actively engage in measures to increase DEI to maximise both participation and the potential for digital innovation | – Businesses should provide teams and team leaders with the resources to enhance their communication skills<br>– Businesses should implement DEI best practices such as developing recruiting strategies, mentoring programs, and personnel training<br>– Funding agencies should fund further research on the needs of under-represented groups specifically affected by the digitalisation of wind energy<br>– Funding agencies should help show the potential for digitalisation by funding open data and benchmarking datasets<br>– Organisations should create sandboxes that can be used for pilot projects |
| Coopetition | – Organisations should support the development of a common core of open-source tools and frameworks which can accelerate innovation and collaboration across the industry | – Businesses and funding agencies should support the creation of common open data standards<br>– Businesses should simplify the process of cooperation by streamlining nondisclosure agreements, licensing, and other nontechnical barriers to cooperation<br>– Governments, sector bodies, or businesses should set up liquid, easily-accessible markets for data and digital services where innovative, collaborative organisations can meet customers<br>– Governments and regulators should encourage transparent public energy data which protects personal privacy whilst facilitating innovation and benchmarking<br>– Businesses and funding agencies should support R&D projects that showcase the use of data marketplaces and public energy data<br>– Organisations should embrace public and private benchmark projects, tools, and datasets to quantitatively show the benefits of digitalisation |



*Author contributions.*   AC led the writing of this paper. AC, SB, AB, PE, JF, AMS, and LW contributed equally to writing and editing all
sections, with the following notable contributions: PE came up with the concept for the first interview series and led the writing of Section
2; AMS proposed the structure for Section 3; AC led Sections 1, 3.1–3.4, 4, and 6; LW led Section 3.5; AB led Section 3.6; SB led Section
3.8; JF and PE led Section 5; SB led Section 5.1.3. YD, JQ, and PT contributed references and to internal reviews. JF led the contribution of
members of IEA Wind Task 43 to interviews and early reviews.

Figure 1 was produced by the NREL graphics team. Figures 2, 3, 4, and 5 were produced by PE and revised by AC. Figure 6 was produced
by AMS. Figure 7 was produced by Josh Bauer, NREL.

*Competing interests.*   AC, SB, YD, PE, JF, AMS, and LW have all received research funding related to the digitalisation of wind energy. AC,
AB, MP, and PT provide commercial services related to the digitalisation of wind energy.

*Disclaimer.*   The mentioning of any company, brand, product, or service in this paper is purely by way of an example and should not be taken
as an endorsement or recommendation.

*Acknowledgements.*   This paper benefited from extensive review by members of IEA Wind Task 43. We thank the reviewers for their contri-
butions. The interviews were conducted by some of the authors and others; the interviewers included Sarah Barber, Anna Maria Sempreviva,
Andrew Bray, Lindy Williams, Jason Fields, Alec Fiala (RES Americas), Yu Ding, Berthold Hahn (Fraunhofer IWES), Des Farren (SERVUS-
Net), Corinne Dubois (Meteolien), and Vasiliki Klonari (WindEurope). We thank the interviewers for their efforts, and the interviewees for
their valuable insights.

Anna Maria Sempreviva and Peter Enevoldsen have received support from the Danish Energy Agency EUDP programme, grant no.
64019-0531.



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
