# Peer review of "Grand Challenges in the Digitalisation of Wind Energy"

_Wind Energy Science, 2022_

## Author Response (AR1)

**WES-2022-29 Reply to reviewers**

Andrew Clifton    Sarah Barber    Andrew Bray    Peter Enevoldsen    Jason Fields
Anna Maria Sempreviva    Lindy Williams    Julian Quick    Mike Purdue
Philip Totaro    Yu Ding

February 2023

**Introduction**

Following is our response to the first reviews of WES-2022-29, "Grand Challenges in the Digitalisation of Wind Energy".

The reviewers' comments are included *like this*. Our responses are interspersed.

We are very grateful to the reviewers for the time they took and effort they expended in reviewing the paper. We thank them for their constructive criticism, which has improved the paper.

**1    Reviewer 1**

The following comments were published as https://doi.org/10.5194/wes-2022-29-RC1.

*Thank you for the opportunity to read this interesting and well-conceived article. I agree with the premise that identifying such Grand Challenges for digitalisation in the wind industry is a valuable exercise and think the authors have done an excellent job of not only reviewing published literature, but reaching out and interviewing individuals, and drawing comparisons to other industries.*

Thank you for these encouraging words!

*My first of my two main comments relates to the Grand Challenges. Their importance is based on the assumption that there is substantial value in digitalisation. Several arguments are made throughout the paper that this is the case, e.g. by comparing to other industries, but as the authors highlight, many digitalisation technologies have existed for some time but are not widely deployed in the wind industry. Is the industry confident that the potential value of digitalisation is large enough to warrant large investments in digitalisation? Do the authors see establishing strong business cases/value propositions for digitalisation as a challenge? Perhaps the value is known to be large enough, in which case this doesn't come across strongly in the paper, but if not, I think the Grand Challenges should reflect this issue.*

This is a good question! The question of assigning a value to digitalisation – and thus, being able to create a business case for it – came up repeatedly in both the interviews and our literature review. We have included this as a more present theme in the new Section 2.

*My second main comment regards the structure of the paper. It is very long compared to other "Grand Challenge" type articles, and the challenges themselves are not introduced until page 30 (other than in the abstract). I think the stated target audience, policy advisers and funding agencies in particular, would more easily digest this work if a shorter exposition of the challenges was presented first, followed by details which they can delve into if desired. I worry that not many would make it to page 30 and miss out on the main outcome of this substantial piece of work.*

We agree that this paper is long compared to other academic papers but would suggest that the length is actually typical for the current "Grand Challenge" papers, and turns into approximately 21 content pages in published form. We have edited this version for length (particularly in Section 1) but note that it is not possible to remove much text without removing context.

At the reviewer's suggestion, the Grand Challenges are now explicitly introduced towards the end of the introduction. To avoid bringing them up without justification we have modified the flow of the introduction to be more of a 'teaser' for the whole paper.

*I also have the following minor comments:*

- *Line 80: The comments around frequency regulation are a little loose. Thermal plant (I think synchronous is more appropriate here) contribute both inertia (through an intrinsic electro-mechanical process) and frequency regulation (through control, e.g. governors). Wind turbines can certainly do the latter and have done so in several power systems for many years, and can provide a response similar to inertia albeit with some delay. I feel that the present phrasing of these issues isn't entirely representative of the present situation. The data need here could be clearer.*

  As suggested in the next comment, we pull back from the specifics and put forward the value of digitalisation as promoting better grid integration broadly.

- *Also ˜Line 80: I might also argue that forecasting and ancillary services are distinct issues and might warrant separating here but I agree they both fall broadly system integration.*

  Similar to above, we pull back from the specifics and put forward the value of digitalisation as promoting better grid integration broadly.

- *Line 85: Is this the same issue as the first bullet in the list?*

  This section was revised to be shorter and combine these points.

- *Line 195: Weird sentence – rephrase?*

  Thank you for pointing this out. This section was eliminated.

- *Line 261-262: Another odd sentence as I read it, perhaps rephrase.*

  Agreed. This section was re-worked to be less wordy and mores simply focus on the cyclical nature of the transition - "both making use of and taking advantage of digitised data and novel data science tools".

- *Line 272-274: Four hyphens make this sentence difficult to read.*

  This sentence and the preceding sentence was re-arranged. We now provide examples in a list.

- *Line 280: SCADA systems certainly simplify data collection, but I think data access depends largely on how the collected data are stored. I've certainly had plenty of experiences where accessing SCADA has been far from simple because of how it has been stored.*

Thank you for pointing this out. We have noted this challenge in the text in Section 4.1. ("Even with access to SCADA data . . . enable data exchange")

- *Line 285: "Lack of access" by whom? Perhaps worth clarifying how different actors have differing abilities regarding access.*

  Thank you. This was not discussed specifically in the first version. We have now clarified who has access to what data (particularly from the turbine SCADA system) in the text in §4.4.

- *Licenses for open data are discussed later but may be worth introducing them around line 425.*

  Agreed. We have added language and citations to this effect.

- *Line 456: Usually there is a trade-off between privacy and accuracy in federated learning which I suggest mentioning here.*

  Agreed. Added language to that effect

- *Line 462: related to my first main comment above, what is this "tremendous value"? Is it there in all cases? This statement should be qualified or supported by some strong sources/citations.*

  This language has been removed.

- *I don't recognise the term ROM in the context of energy forecasting, perhaps some forecasters use ROMs but they are not that common, or you would call the type of power-curve model forecasters use a ROM. In any case, it is worth noting that some forecasters are moving towards use of high-performance computing to run ever higher resolution atmospheric models for real-time applications rather than scaling back on complexity.*

  Reduced Order Models (ROMs) are also known as an engineering models and are a commonly accepted approach to time or cost sensitive applications that may not be suitable for high performance computing applications. We have modified the text in §4.5 to provide more background and introduce the name.

- *"Virtuous upward spiral". Thank you for introducing me to this fun term. I had to look it up and presume that here your intended meaning is along the lines of "a cycle of compounding successes". Can the claim in this sentence be supported by a citation?*

  We have lightly updated this sentence and added several citations that support the assertion that digitalisation can enable valuable collaboration with customers and partners.

- *Line 501: "One challenge with digitalisation is that there is an adoption process." Surely the challenge is due to some properties of the adoption process, not the existence of one.*

  Agreed. This paragraph has been revised to be clearer about the nature of the adoption challenge.

- *Line 618: According to the data in the link, $318bn revenue was generated by smartphone apps in 2020, not 582. Online sources should be properly referenced with a date of access. The same applies to footnotes 6 and 7, which are also not aligned with information currently at the end of those links.*

  Thank you for pointing this out. We have updated our references.

- *Section 5.1.1: Is the use of subsubsection necessary? They are only used under subsection 5.1, I think.*

  Subsubsections have now been removed.

- *Line 693: I don't think the meaning of this bullet is clear.*

  This bullet has been fixed. The other bullet points describing the challenges have also been updated.

- *Line 703: Please check the definition of "diversity".*

  Done. The definition has been lightly updated.

- *Line 711: What is "the population"?*

  Done. Changed to "the general population"

- *Line 882: Should this be FAIR data frameworks rather than "reusable"?*

  Yes. We have updated this.

- *The interview and survey data should be FAIR and attached to this article if at all possible.*

  We have attached the interview questions from IEA Wind Task 43. However the individual interview responses are confidential and the participants only agreed to make aggregated results available.

**2    Reviewer 2**

The following comments were published as https://doi.org/10.5194/wes-2022-29-RC2.

*This paper presents, in a concise and easy-to-read way, the various aspects and challenges of digitalisation in the wind industry. The ideas are clear and the writing is argumentative.*

*Overall, it is a very interesting piece of work, worth the attention of the scientific community besides the targetted audience (policy advisers and funding agencies). I recommend this paper for publication, subject to only minor revision - please refer to the comments and feedback below.*

Thank you for the positive feedback. In light of your comments and reviewer 1's comments we have made a few changes to the manuscript for readability and focus. These are discussed in context below.

*General Comments:*

- *In Section 1.2, the authors provide a list of opportunities for the collection and use of data in the wind industry. Although comprehensive, this list is used only to gather examples of applications. I believe it would be valuable to group the opportunities here listed into classes, based on several collection purposes. For instance, I could identify: (i) site assessment for design, (ii) condition and health monitoring (diagnosis and prognosis) for operation and maintenance, (iii) control and electrical engineering for grid integration, (iv) wind resource assessment and array aerodynamics for reliability assessment, performance evaluation and optimisation, (v) all above and any other type of data integrated into virtual/augmented reality for HSE.*

  Section 1 has been extensively restructured and the opportunities that the reviewer referenced have been largely rolled into a new Section 2, where we discuss the potential implications of digitalisation for LCOE.

- *I would recommend moving some of the paragraphs in Section 1.2 (line 93-103) and 1.3 (line 148-154) to Section 1.5, by remaining it into "Scope and objectives of this Grand Challenge for digitalisation". There, the authors explain why the path towards the digitalisation of the wind industry is challenging but essential. These paragraphs set the ground for the assessment performed in the paper, and they would fit best in Section 1.5.*

  Thank you for the comment. Please note our response to your previous comment. We agree that a restructuring was required to make the paper easier to follow.

- *I would suggest moving Section 1.4 to an appendix, by only briefly referring to it in Section 1.5. Although I found it an interesting and innovative way to engage the reader with the story, this section distracts a bit from the introduction to this research. On the other hand, it can be linked easily to the goal of this paper, which is to identify the challenges and enablers to make the story come true.*

  We have elected to remove the story in Section 1.4 as it was also considered distracting in internal reviews.

- *On the points raised in the first paragraphs of Section 3.4, I agree with the need for wind-wise and across-industries (power generation, at least) taxonomies. For this reason, the RDS-PP (Reference Designation System for Power Plants), and more recently the RDS-PS (... Power System), standards were created. However, the main issues still lie in their accessibility and access (quite pricey).*

A reference to the RDS-PP has been added. We agree that there are challenges with the current approach to closed, pay-to-read standards. These are similar to the challenges facing scientific publishing and we suspect that similar "open science" and "open access" initiatives may affect standards in the next decade. We felt though that this discussion is out of the scope of the paper and so have not included it.

*Some other minor comments:*

- *Line 107: Is it possible to have a reference on these 1% savings of the CAPEX? What type of studies were performed (and by who) to state this?*

  This text has been removed. Instead we note that digitalisation will be a contributor to the expected / hoped-for 35-40% reduction in LCOE by 2050.

- *Line 110-112: I would add that digital-enabled asset management, allowing to implement of a condition-based maintenance strategy, also has the potential to reduce and/or challenge the number of recommended inspections and schedule maintenance tasks (see for instance deliverable 4.3 of the COREWIND project, soon to be published open access at https://corewind.eu/publications/)*

  This potential application has been included in the discussion in Section 2.

- *Line 124-128: I agree with the authors about the need to quantify the potential ROI of the digital services and technologies, and about the current lack of industry-proven cost figures for such investment. However, some researchers have tried to provide some first estimates (e.g. https://www.sciencedirect.com/science/article/pii/S0951832020308905)*

  Thank you for this suggestion. We have noted in Section 2 that cost models (such as Koukoura et al.) could be used to test certain digitalisation strategies, but cannot be applied to the whole sector.

  We also note that we expect that digitalisation will be one of the enablers of the 20-40% reduction in LCOE by 2050 forecast by the expert solicitation reported by Wiser et al.

  Given the huge number of potential applications for digital services in wind energy, we expect that value propositions and ROI will have to be identified by service providers on a case-by-case basis or for each product.

- *Line 144-147: Is there an "and" too much? Please rephrase and/or consider presenting as bullet point (no need for numbered list)*

  This section has been revised and this text no longer exists.

- *Figure 2: "Data collection, processing, and analysis were used to help identify..." could be replaced by "Flowchart of..."*

  Thank you for this suggestion. The caption has been updated.

- *Line 220-221: "The results..." repeats twice "the results of the literature survey" – correct?*

  Thank you for pointing this out. This has been corrected. This has been corrected.

- *Line 272-274: "And, processes ..." is difficult to read, can you please rephrase it?*

  This paragraph has been rewritten and a few examples of digitisation are now included as bullet points.

**END**